# Enabling valley selective exciton scattering in monolayer $WSe_2$ through upconversion

M. Manca[1], M.M. Glazov[2], C. Robert[1], F. Cadiz[1], T. Taniguchi[3], K. Watanabe[3], E. Courtade[1], T. Amand[1], P. Renucci[1], X. Marie[1], G. Wang[1] & B. Urbaszek[1]

Excitons, Coulomb bound electron–hole pairs, are composite bosons and their interactions in traditional semiconductors lead to condensation and light amplification. The much stronger Coulomb interaction in transition metal dichalcogenides such as $WSe_2$ monolayers combined with the presence of the valley degree of freedom is expected to provide new opportunities for controlling excitonic effects. But so far the bosonic character of exciton scattering processes remains largely unexplored in these two-dimensional materials. Here we show that scattering between B-excitons and A-excitons preferably happens within the same valley in momentum space. This leads to power dependent, negative polarization of the hot B-exciton emission. We use a selective upconversion technique for efficient generation of B-excitons in the presence of resonantly excited A-excitons at lower energy; we also observe the excited A-excitons state 2s. Detuning of the continuous wave, low-power laser excitation outside the A-exciton resonance (with a full width at half maximum of 4 meV) results in vanishing upconversion signal.

[1] Université de Toulouse, INSA-CNRS-UPS, LPCNO, 135 Av. Rangueil, Toulouse 31077, France. [2] Ioffe Institute, St Petersburg 194021, Russia. [3] National Institute for Materials Science, Tsukuba, Ibaraki 305-0044, Japan. Correspondence and requests for materials should be addressed to M.M.G. (email: glazov@coherent.ioffe.ru), G.W. (email: g_wang@insa-toulouse.fr) or to B.U. (email: urbaszek@insa-toulouse.fr).

In semiconductors the optical properties are governed by excitons, bound electron–hole pairs[1–6] with certain analogies to the hydrogen atom. The exciton energy states and polarization selection rules need to be understood for designing optoelectronic applications which target efficient emission and strong light–matter interaction[7]. Excitons also provide a rich platform for fundamental physics experiments[8–13]. A real breakthrough for exciton physics in the solid state was to show that excitons behave like composite bosons[14–16]. The key effect is stimulated scattering: Bosons preferentially scatter to a quantum state that is already occupied. These pioneering works enabled studies into optical amplifiers based on excitons and also new fundamental research on exciton Bose–Einstein condensation, which are still ongoing[17–22].

Our target in this work is to look for fingerprints of bosonic interactions in two-dimensional (2D) materials. Excitons in transition metal dichalcogenide (TMDC) monolayers (MLs) provide exciting new opportunities for applications and new frontiers in exciton physics for several reasons: First, with binding energies of several hundred meV (refs 23–29), excitons dominate optical properties even at room temperature. The corresponding exciton Bohr radius is of the order of only 1 nm, leading to a limit for the Mott-transition at much higher densities than in conventional semiconductors, allowing to explore a wider exciton density regime. Second, the strong exciton oscillator strength leads to absorption of up to 20% per monolayer[30,31], and third, the interband selection rules are valley selective. In combination with strong spin-orbit splittings this allows studying spin-valley physics[32–37]. These unique excitonic properties make ML TMDCs ideal systems for investigating exciton interactions[38–43] and microcavity polariton physics[44–54].

We introduce an original optical excitation scheme to study exciton scattering in the model 2D TMDC monolayer material WSe$_2$. We provide a unique situation for efficient generation of B-excitons in the presence of resonantly excited A-excitons at lower energy. We resonantly excite the A-exciton with a low-power laser using an extremely high-quality sample with only 4 meV full width at half maximum (FWHM) transition linewidth. Surprisingly, we observe emission from the B-exciton, 430 meV above the A-exciton state and also of the excited A-exciton (2s) 130 meV above the fundamental, 1s, state, which we refer to for brevity as upconversion photoluminescence (PL). For circularly polarized excitation we show that the upconverted B-exciton emission is strongly cross-circularly polarized, the polarization degree increases with laser excitation power. This can be interpreted as a first fingerprint of boson scattering of 2D excitons[14–16] that favours relaxation from the B- to A-excitons within the same valley in momentum space. Possible mechanisms of the upconversion in ML WSe$_2$ are discussed[55] and compared to upconversion reported for more traditional nanostructures such as InP/InAs hetero-junctions, CdTe quantum wells and InAs quantum dots[56–60].

## Results

**Upconversion emission 430 meV above excitation laser.** We study WSe$_2$ MLs encapsulated in hexagonal boron nitride (h-BN)[61]. The aim is to eliminate detrimental surface effects[43] and to provide a symmetric (top/bottom) dielectric environment to study excitons. The high-optical quality of the sample is demonstrated in Fig. 1c: here we show reflectivity spectra using a white light source for illumination. We detect the A exciton peak at 1.723 eV, with a linewidth of typically 4 meV, this main exciton transition is labelled 1s in analogy to the hydrogenic model. Although the A:1s energy in uncapped ML WSe$_2$ on SiO$_2$ is very similar at 1.75 eV, the impact of the change in dielectric

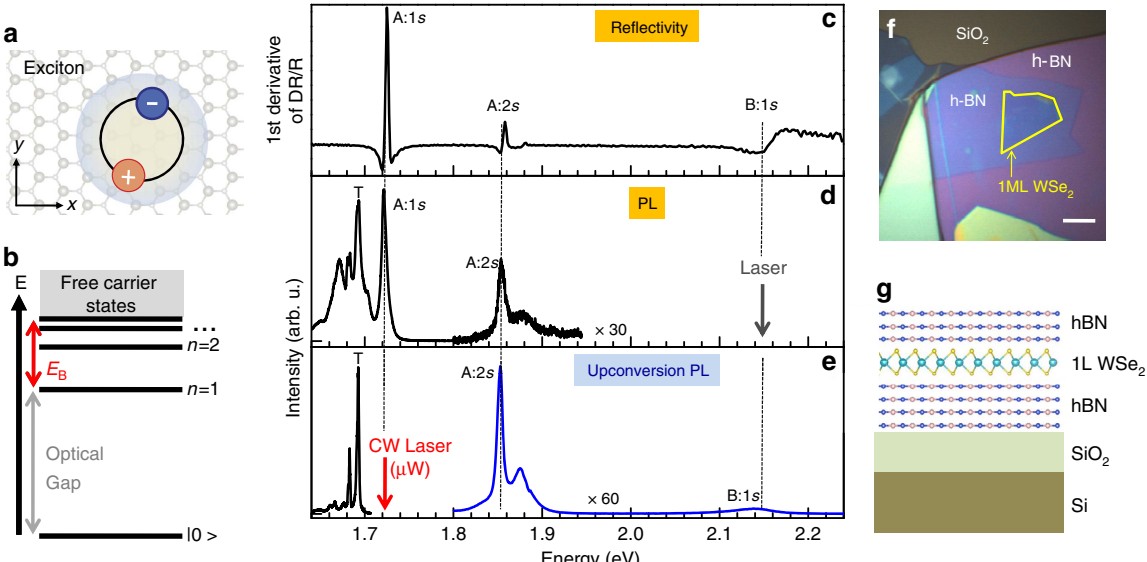

**Figure 1 | Exciton resonances in linear spectroscopy and upconversion at $T = 4$ K.** The sample consists of a WSe$_2$ monolayer encapsulated in hBN (ref. 61). (**a**) Strongly bound electron–hole pairs, excitons, dominate optical properties of TMDC monolayers such as WSe$_2$. In WSe$_2$ there exist two different exciton series. The B-exciton is about 400 meV above the A-exciton due to spin-obit splitting of the valence band. (**b**) Excitons have a binding energy $E_B$, defined as the difference between the free particle bandgap and the optical bandgap observed in photoluminescence (PL) emission. $E_B$ is of the order of 500 meV, the first excited state $n = 2$ is about 130 meV above the $n = 1$ state, marked as A:2s and A:1s, respectively, throughout this manuscript. (**c**) We have performed reflectivity with a white light source to identify the A- and B-exciton at $T = 4$ K. In addition we observe an excited states of the A-exciton labelled A:2s. (**d**) In photoluminescence we observe neutral A-exciton and trion emission (T), in addition we see hot luminescence of the A:2s state. This is the only spectrum obtained using a pulsed laser in this work, to reach the energy of the B:1s, see Methods. (**e**) We also demonstrate upconversion PL: the laser is tuned to the A:1s-exciton resonance and strong emission from B:1s and A:2s at much higher energy is recorded, in addition to the trion emission (T) at lower energy. (**f**) optical microscope image of the studied van der Waals heterostructure, with scale-bar of 5 μm. (**g**) Scheme of the side view of the sample.

environment through encapsulation is not yet well understood. Recent experiments[62] suggest that although the exciton energies A:1s are similar in both types of samples, this might be the result of a decrease in exciton binding energy, that is almost entirely compensated by a decrease in the free carrier bandgap. About 133 meV above the A:1s we detect an excited exciton state, as demonstrated before for samples exfoliated onto $SiO_2$ (refs 26,27,63,64). The linewidth of this transition, which we tentatively ascribe to the 2s A exciton, is of the order of 5 meV. It is accompanied by a smaller peak about 25 meV at higher energy which could be related to the A:3s exciton. At 430 meV above the A:1s we detect the B-exciton transition, where the A–B separation is mainly given by the valence band spin-orbit splitting[65].

In addition to white light reflectivity, we have also performed PL experiments shown in Fig. 1d. Using an excitation laser energy resonant with the B-exciton transition, we observe in addition to the A:1s exciton emission (4 meV FWHM) and the trion (T) also the hot A:2s emission (5 meV FWHM), at very similar energies as the reflectivity results, indicating negligible Stokes shifts. It is a signature of the high-quality of our sample, since the absence of the Stokes shifts indicates only very weak localization of excitons, if any. The trion emission is detected in PL but not clearly in reflectivity, which indicates a lower resident electron concentration as compared to the neutral exciton[66,67]. In our samples the

biexciton emission from a molecule-like two exciton complex with a characteristic superlinear intensity dependence on the incident laser power has not been observed. Hot PL of the A:2s is also observed for other laser energies such as 1.96 eV (HeNe Laser).

The optical spectra shown in Fig. 1e, introduce our upconversion scheme: excitation of the A:1s exciton ($E_L = 1.723$ eV) results in strong PL emission at higher energy of the A:2s ($E_L + 133$ meV) and B:1s exciton ($E_L + 430$ meV), as well as trion emission (labelled T) at lower energy $E_L - 31$ meV. This upconversion is achieved using a narrow linewidth, continuous wave laser and moderate excitation powers in the $\mu W \cdot \mu m^{-2}$ range, see Methods. Interestingly we also observe in WSe$_2$ MLs directly exfoliated onto $SiO_2$ (no hBN in the structure) this upconversion emission for A:2s and B:1s, see Supplementary Fig. 1 and Supplementary Note 1. These observations are very surprising and further experiments that aim to clarify the origin of this upconversion are shown in Fig. 2.

**Investigating the origin of the upconversion signal.** The observed upconversion is extremely energy dependent: only strictly resonant excitation of the A:1s exciton results in measurable upconversion luminescence. The FWHM of the observed resonance in upconversion PL excitation (PLE) is about 4 meV, as

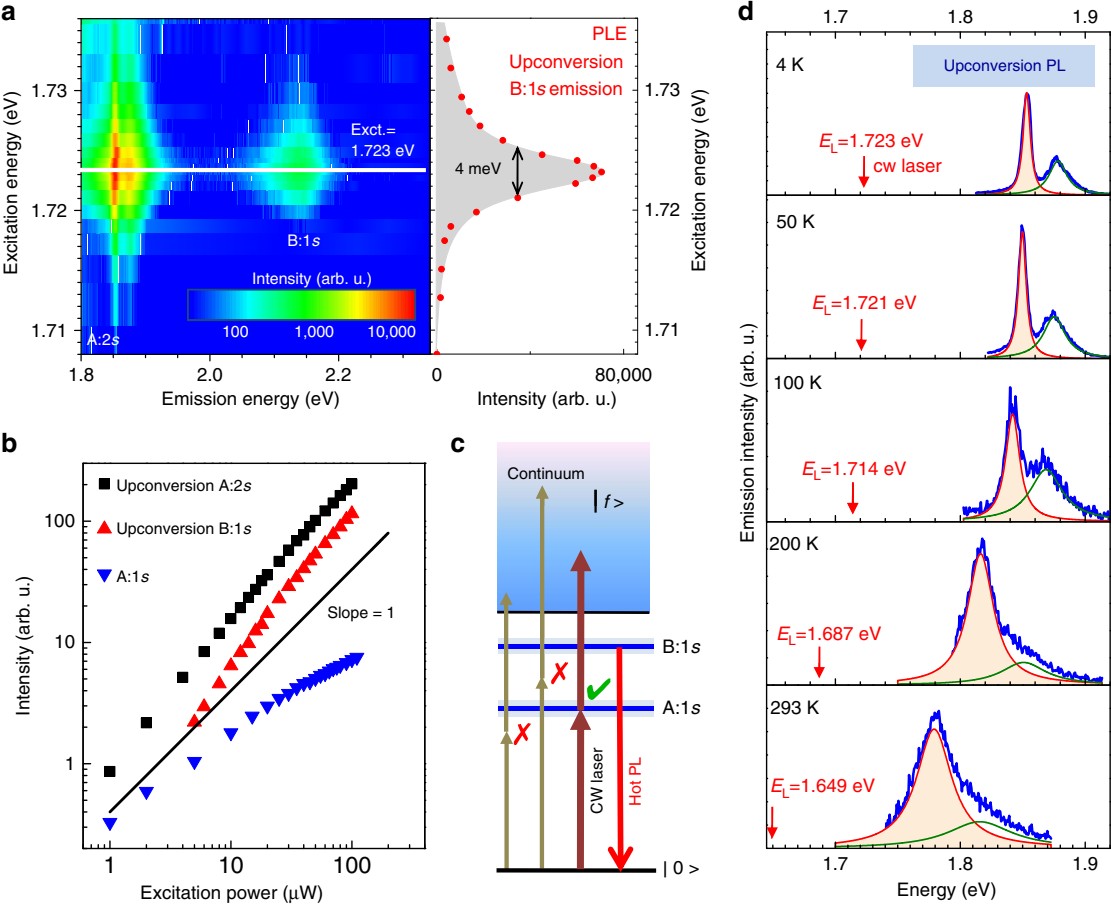

**Figure 2 | Investigating the origin of the upconversion process.** Sample temperature $T = 4$ K. (**a**) Left panel: contour plot of the upconversion PL intensity for A:2s and B:1s as a function of excitation energy, with a clear resonance at 1.723 eV, the A:1s exciton transition energy. Right panel: The resonance in upconversion PLE has a FWHM of only 4 meV. (**b**) We show power dependence of the upconversion PL intensity at resonance and compare with power dependence of the neutral exciton excited at the A:2s state in standard PL. Slopes in the range of 10-100 μW are for A:2s = 1.1, B:1s = 1.25 and A:1s = 0.56 (**c**) Scenario for upconversion PL based on 2-photon absorption aided by a real intermediate state is presented. (**d**) Evolution of upconversion PL of A:2s for $T = 4$ K up to room temperature; the excitation laser energy is indicated following the shift of the A:1s exciton resonance with temperature.

shown in Fig. 2a, see red data points. Resonant excitation 31 meV below the A:1s state at the trion energy for example[55], does not result in emission at the A:2s and B:1s energies in our experiment. Emission at higher energy than the laser can have several origins[56–60,68], here important information comes from the B-exciton emission: at 430 meV above the laser energy mechanisms purely based on phonon emission (that is, laser cooling[69]) are very unlikely at the sample temperature $T = 4$ K. A more probable scenario is two photon absorption, made efficient by the A:1s as a real intermediate state. This idea is supported by the narrow resonance around the A:1s exciton (Fig. 2a) and also by analysis of the power dependence of the emission: the upconversion PL evolves with a slope roughly twice that of the exciton A:1s emission, see Fig. 2b. This indicates that two excitons resonantly excited by the laser combine to form a single excited state of the electron–hole pair, with the energy being the sum of exciton energies. As a plausible scenario we may suggest an Auger-like process, also referred to as exciton annihilation[38–41]. In this case the scattering of two existing excitons results in the transition of one electron forming an exciton to the valence band (that is, nonradiative recombination[70]), while the remaining electron absorbs the released energy and is promoted to the excited energy band denoted as a continuum $|f\rangle$ in Fig. 2c. Subsequently, the excited electron–hole pairs loose energy via, for example, phonon emission and relax towards the radiating states, particularly, B:1s and A:2s. As a result, the upconversion intensity scales as $N_{A:1s}^2$, where $N_{A:1s}$ is the exciton occupancy created by the laser. The occupancy $N_{A:1s}$ is directly proportional to the intensity of the exciton emission from A:1s state, in agreement with experiment, see also the detailed discussion in Supplementary Note 2, using Supplementary Figs 5 and 6 and Supplementary Table 1. In addition we observe that upconversion PL in our sample is detectable for the A:2s state even at room temperature, see evolution as a function of temperature in Fig. 2d and more detail in Supplementary Fig. 4.

**Possibility of boson scattering from B- to A-exciton levels.** In our experiment we resonantly pump the A-exciton transition. As addressed above, presumably an additional photon is absorbed to create a second exciton and, ultimately, to generate an electron–hole pair in a high-energy continuum state. From there the electron–hole pairs relax towards the B-exciton, where we observe hot luminescence 430 meV above the A-exciton. Very surprisingly the B:1s emissions is strongly $\sigma^-$ polarized following circularly polarized $\sigma^+$ excitation, that is, counter-polarized with respect to the laser, Fig. 3a–c. A strong indication for the importance of scattering processes comes from power dependent measurements: for a laser excitation power of 5 μW we measure $-2 \pm 2\%$ PL polarization, for a stronger excitation power of 50 μW the negative polarization is about $-14 \pm 2\%$, see Fig. 3a–c. Observing any polarization at all for these upconversion PL signals is extremely surprising, as it is expected that absorption at

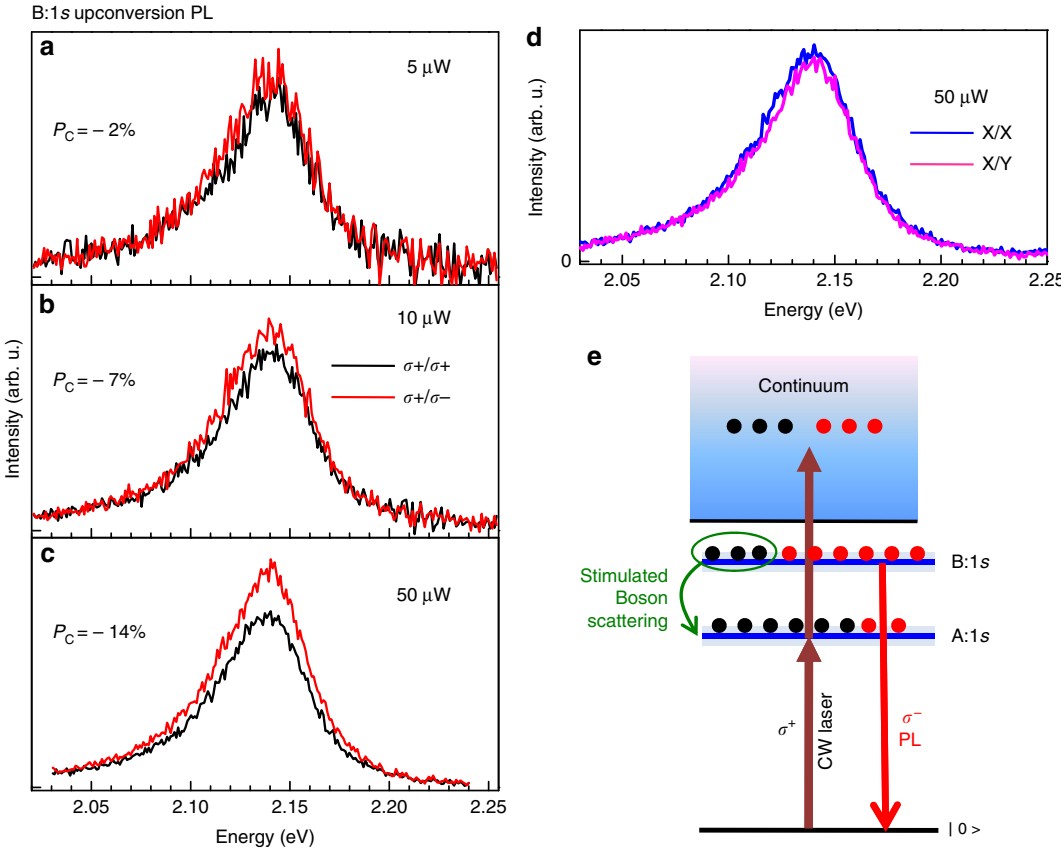

**Figure 3 | Creating B- and A-excitons and first signatures of boson scattering.** The sample temperature is $T = 4$ K, $E_L = 1.723$ eV, $\sigma^+$ laser polarization. (**a**) B:1s upconversion PL detected in $\sigma^+$ (black—co-polarized) and $\sigma^-$ (red-cross-polarized) polarization for a laser power of 5 μW. (**b**) Same as (**a**), but for 10 μW. (**c**) Same as (**a**) but for 50 μW. The value of the circular polarization degree of the emission $P_c = \frac{I^{\sigma+} - I^{\sigma-}}{I^{\sigma+} + I^{\sigma-}}$ is indicated on the panel. As we increase the power, the emission becomes strongly cross-polarized with respect to the initial excitation, we generate a negative polarization $P_c < 0$. (**d**) Laser excitation linearly ($X$) polarized, detection of upconverted PL at B:1s energy in linear $X$ and $Y$ basis. (**e**) Scheme to explain negative, power depedent polarization of hot B:1s PL emission observed in **a–c** based on boson scattering.

high-energy continuum states as well as Auger-type processes are not governed by strict polarization selection rules. In this context we have verified that linearly polarized excitation and circularly polarized excitation yield exactly the same intensity of the upconversion signal[71].

Next we aim to explain the strong, negative polarization of the B:1s emission. We proposed a scenario based on stimulated boson scattering[14–16], as sketched in Fig. 3e. As a first step, we assume excitation creates an equal population of $\sigma^+$ and $\sigma^-$ excitons in the continuum $|f\rangle$, as chiral selection rules are relaxed. Subsequently the electron–hole pairs relax towards the B:1s state with the same rates for $\sigma^+$ and $\sigma^-$. However, the relaxation from the B-excitons towards the ground, A:1s states is polarization-dependent as bosons preferentially scatter to a quantum state that is already occupied: the pump laser creates a majority of co-polarized A-excitons and since excitons are bosons, the scattering probability of $\sigma^+/\sigma^-$ polarized B:1s excitons towards A:1s excitons grows as $(1 + N_{A:1s}^{\sigma\pm})$, where $N_{A:1s}^{\sigma\pm}$ are the occupancies of correspondingly polarized A:1s-excitons (see ref. 16). As a result, co-polarized B-exciton states get depleted faster than counter-polarized B-excitons. This imbalance gives rise to hot B-exciton PL emission counter-polarized with respect to the excitation laser. Linearly polarized excitation of A:1s does not induce any linearly polarized upconversion emission of B:1s, see Fig. 3d. Linear polarization is linked to valley coherence[34], which is too fragile to be maintained during the upconversion and energy relaxation processes.

**Anti-Stokes Raman scattering**. The upconversion PL of the A:2s transition on the other hand is strongly co-polarized with the excitation laser with $P_c \approx 25\%$, here the dependence on excitation laser power is rather weak, as shown in Fig. 4a–c. We argue that the polarization of the A:2s is similar to the A:1s polarization as the exciton populations of the two states are coupled. First, the A:2s to A:1s separation is only 133 meV, compared to the B:1s to A:1s separation of 430 meV. Second, we observe anti-Stokes Raman scattering superimposed on the hot PL of the A:2s exciton, as can be seen in Fig. 4e; Supplementary Fig. 2. Previously, we have reported double resonant Stokes Raman scattering[63], that showed efficient relaxation from the A:2s state to the A:1s state as they are separated by a phonon-multiple. Here the equivalent anti-Stokes process is visible in the experiments. Due to efficient phonon exchange between the A:1s and A:2s states the polarizations of the ground and excited states have the same sign. Note that we cannot probe the A:1s polarization directly in resonant excitation conditions (signal is obscured by scattered laser light). In these experiments at $T = 4$ K the phonons can be generated by the relaxation following two photon absorption, for example, as well as due to the exciton to trion conversion through phonon emission[55]. Just as for the B:1 s upconversion, the experiments using linearly polarized lasers do not result in linearly polarized emission in Fig. 4d.

## Discussion

In summary, we demonstrate upconversion photoluminescence in WSe$_2$ monolayers at energies as high as 430 meV above the laser energy. The effect occurs for strictly resonant excitation of the ground A-exciton state 1s and is most probably related to two photon absorption enabled by a real intermediate state. Very surprisingly, the upconverted PL emission of the B-exciton is counter-circularly polarized with respect to the excitation laser, which provides a fingerprint of stimulated exciton scattering from B- to A-states, which efficiently depletes the co-polarized B-exciton state. In future experiments it would be interesting to see how the order of dark versus bright exciton states impacts this

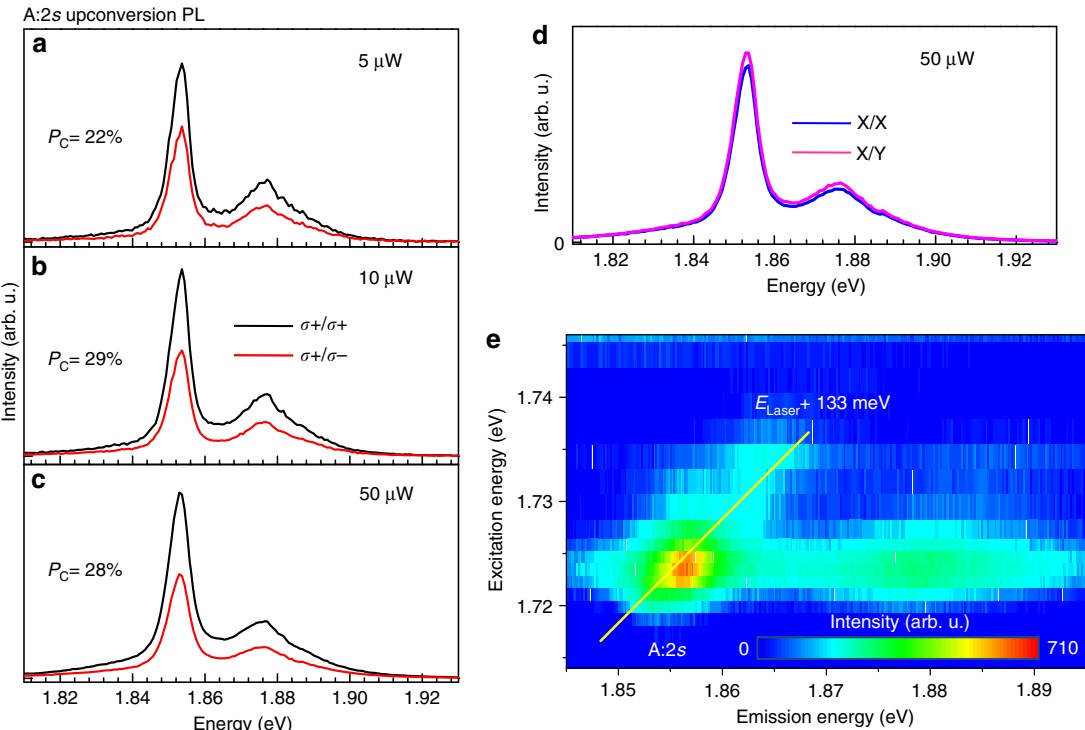

**Figure 4 | Photon emission 133 meV above the excitation laser at 4 Kelvin.** (**a**) $T = 4$ K, $E_L = 1.723$ eV $\sigma^+$ laser polarization. Upconversion PL of A:2s state detected in $\sigma^+$ (black) and $\sigma^-$ (red) polarization for a laser power of 5 μW. (**b**) Same as (**a**), but for 10 μW. (**c**) Same as (**a**) but for 50 μW. (**d**) Laser excitation linearly (X) polarized, detection of upconverted PL at A:2s energy in linear X and Y basis. (**e**) Contour plot of A:2s upconversion PL as a function of laser energy, a Raman feature moving with the laser energy is clearly visible, see Supplementary Figs 2 and 3 for water-fall style plot.

scattering mechanism, as for example in ML MoSe$_2$ the lowest lying transition is the bright exciton, contrary to ML WSe$_2$. We also show upconversion emission at an energy corresponding to the excited state of the A-exciton 2s. Here strong phonon effects are visible in the form of anti-Stokes emission which exactly shifts with the excitation laser energy $E_L$ as $E_L + 133$ meV.

## Methods

**Samples.** The WSe$_2$ ML flakes are prepared by micro-mechanical cleavage of a bulk crystal (from 2D Semiconductors) and deposited using a dry-stamping technique on hexagonal boron nitride[61] on SiO$_2$/Si substrates. Subsequently h-BN was deposited on top of the WSe$_2$. Figure 1f shows an optical microsope image of the fabricated van der Waals heterostructure.

**Experimental set-up.** Low temperature PL and reflectance measurements were performed in a home build micro-spectroscopy set-up build around a closed-cycle, low vibration attoDry cryostat with a temperature controller ($T = 4$–300 K). For PL at a fixed wavelength of 633 nm a HeNe laser was used, for PL experiments as a function of excitation laser wavelength we used a tunable, continuous wave Ti-Sa Laser SolsTis from M SQUARED allowing continuous tuning in the range of 700–1,000 nm. For wavelength below 700 nm in Fig. 1d the sample is excited by picosecond pulses generated by a tunable frequency-doubled optical parametric oscillator (OPO) synchronously pumped by a mode-locked Ti:Sa laser. The typical pulse and spectral width are 1.6 ps and 3 meV, respectively; the repetition rate is 80 MHz (ref. 72). The white light source for reflectivity is a halogen lamp with a stabilized power supply. The emitted and/or reflected light was dispersed in a spectrometer and detected by a Si-CCD camera. The excitation/detection spot diameter is $\approx 1$ µm, that is, smaller than the typical ML diameter.

**Data availability.** The data that support the findings of this study are available from the corresponding author on request.

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

## Acknowledgements

We thank ERC grant no. 306719, ITN Spin-NANO Marie Sklodowska-Curie grant agreement no. 676108, ANR MoS2ValleyControl, Programme Investissements d Avenir ANR-11-IDEX-0002-02, reference ANR-10- LABX-0037-NEXT and LIA CNRS-Ioffe RAS ILNACS for financial support. X.M. also acknowledges the Institut Universitaire de France. M.M.G. is grateful to RFBR, Russian Federation President grant MD-1555.2017.2, and Dynasty Foundation for partial support. K.W. and T.T. acknowledge support from the Elemental Strategy Initiative conducted by the MEXT, Japan and JSPS KAKENHI grant numbers JP26248061,JP15K21722 and JP25106006. We thank Alexey Chernikov and E.L. Ivchenko for very fruitful discussions.

## Author contributions

M.M., G.W., F.C. and E.C. performed the measurements. K.W. and T.T. grew the hBN. C.R. and P.R. fabricated and tested the van der Waals heterostructures. M.M.G, T.A., X.M. and B.U. interpreted the data, B.U., G.W. and M.M.G. wrote the manuscript with input from all the authors.

## Additional information

**Competing interests:** The authors declare no competing financial interests.

**Publisher's note**: 

