## [Peer Review File · Nature Communications]

Reviewers' Comments:

Reviewer #1 (Remarks to the Author)

The authors demonstrate upconversion of photoluminescence (PL) in WSe₂ monolayers when they are excited at resonance with the A-exciton 1s state. The upconversion gives rise to B-exciton PL. This effect is presumably due to a two-photon absorption with a real intermediate state (the A-exciton state). The observation the authors find the most surprising is the opposite helicity of the upconverted PL. They argue that it is the evidence for stimulated (nonradiative) exciton scattering from B- to A-states.

In terms of significance and potential impact, I find this work somewhere in the middle. The quality of the data is impressive, especially the narrow exciton linewidths. The upconversion effect is also interesting. However, this effect is probably of most interest to researchers in the specific field of TMD optics. I do not see it to be super-exciting for broader physics audience. As for the claim of stimulated scattering, it is plausible but lacks a solid proof. The authors did not demonstrate any control over this putative process.

I am undecided regarding whether this paper deserves publication in Nature Comm. In my opinion, it is a marginal case.

My final comment is about missing citations. In the 1st paragraph, the grouping [8-12] includes the follow-up work of Ref. 10 but not the original discovery [Butov et al, Nature 418, 751 (2002)]. In the second grouping [16-20] an important paper on exciton condensation [High et al, Nature 483, 584 (2012)] is overlooked. In the 2nd paragraph the grouping [42-52] includes Ref. 49 but not the earlier competing publication: Basov et al, Science 354, no. 6309 (2016), DOI: 10.1126/science.aag1992 .

Reviewer #2 (Remarks to the Author)

The authors report on low-temperature optical spectroscopy experiments on a monolayer of the TMDC WSe₂. In measurements where they illuminate the A exciton transition in their sample resonantly, they observe luminescence emission peaks at energies well above the laser energy, and identify these high-energy peaks as excited A exciton states and B exciton states. They demonstrate a strong sensitivity of this 'upconverted' luminescence on excitation laser energy, and a power dependence of the luminescence intensity which has about twice the slope of the 1s A exciton luminescence. The authors propose a mechanism of 2-photon absorption, with the 1s A exciton as a real intermediate state, leading to formation of B excitons from highly excited (continuum) states.

By using circularly polarized, resonant excitation of the 1s A exciton transition, they are able to generate significant circular polarization of the upconverted, excited A exciton state and B exciton emission. Remarkably, the circular polarization degree of these two luminescence peaks is opposite to each other, and the authors attribute the counter-polarized B exciton emission to B-exciton-A-exciton scattering, which is more probably between excitons in the same valley. This mechanism would lead to an imbalance of B exciton occupation in the different valleys, even though the B exciton formation is not valley-index-conserving. This mechanism is supported by density-dependent measurements clearly showing increasing counter-polarized luminescence due to increased exciton-exciton scattering.

The results presented in this manuscript are novel, and the claims made by the authors are well-supported by the experiments. They convincingly present an elegant way to establish a B exciton valley polarization in a TMDC, and their results are likely to invigorate further research of B exciton valley dynamics. More importantly, they provide clear evidence for valley-selectivity of bosonic scattering processes in a TMDC, which will have to be considered in future research aimed at

studying exciton-exciton interaction and potential exciton condensation. Thus, the paper will most certainly be highly interesting to other groups working in this field.

Overall, the paper is very well-written, and the figures are clear and informative. While I would highly recommend publication of this manuscript in Nature Communications without major changes, I do have some questions which should be addressed by the authors, as well as some suggestions for minor revisions of the paper.

My questions are related to phenomena which are, so far, specific to the tungsten-based TMDCs: the conduction-band spin splitting in these materials makes the lowest-energy A exciton state a dark state, as demonstrated, among others, by some of the authors of the present work (NATURE COMMUNICATIONS | 6:10110). By contrast, the lowest-energy B exciton state (with the electron in the lower spin-split conduction band) is bright. Have the authors considered the potential impact of this splitting on the effects they observe? Might this actually limit the feasibility of the B exciton generation mechanism in the related TMDC MoSe₂?

The second question concerns the formation of biexcitons, which have been reported for WSe₂ under intense excitation. Do the authors have any indications of biexciton emission for their highly resonant A exciton excitation conditions?

Suggestions for minor changes:

- There is an additional feature above the 2s A exciton transition that is tentatively identified as the 3s A exciton – is the energy difference in agreement with the models for the excited states of TMDC monolayers?
- Related to the point above: a two-component fit is used to track the 2s A exciton and the higher-energy feature as a function of temperature, but this is not discussed in the caption of Fig.2d, and should be mentioned there. It also appears that there are significant changes of the linewidth, spectral weight and maybe also of the splitting between the two features. Given that these features have already been analyzed, it might be helpful to include the data from these fits in the supplement for interested readers.

Small comments:

- The authors mention the use of a pulsed laser in the methods. I assume that this was used to resonantly excite the B exciton transition for the PL spectra shown in Fig.1d? If that is the case, it should be mentioned in the caption of the Figure or in the text, or added within the figure itself. I would also suggest to indicate for which other measurements the pulsed excitation was employed,
- While the continuum states $|f\rangle$ are mentioned in reference to Fig. 2c both within the main text and the supplementary, the actual symbol $|f\rangle$ is not depicted in that figure.

Reviewer #4 (Remarks to the Author)

This paper delivers interesting and important result for the highly competitive field of TMDC materials, where the efforts of many leading groups in solid state physics are focused nowadays. The authors present very nontrivial experimental observation of the strong upconversion signal from B-exciton state, when the excitation is resonant for the A-exciton, which is lower in energy. Even more surprising is the fact that the emission of B-exciton is circularly polarized when the circular polarized excitation is chosen, and the polarization degree is negative to the excitation one. By careful experimental checks the authors convincingly exclude few possible mechanisms and conclude that this polarization originates from the bosonic character of the exciton scattering. This is novel and principally important result for TMDC, other 2D materials, but also semiconductor microcavities.

Paper is very clearly written to be understandable for general reader. I recommend it for publication in Nature Communications.

I have only one question that author may wish to comment. Does encapsulation of WS₂ in boron nitride influence of the exciton parameters (e.g. binding energy), say via dielectric confinement?

Reply to reviewers

Below we answer the reviewers' questions point by point. The resulting changes, where applicable, appear in blue font in the revised manuscript text.

Reviewer #1 (Remarks to the Author):

The authors demonstrate upconversion of photoluminescence (PL) in WSe₂ monolayers when they are excited at resonance with the A-exciton 1s state. The upconversion gives rise to B-exciton PL. This effect is presumably due to a two-photon absorption with a real intermediate state (the A-exciton state). The observation the authors find the most surprising is the opposite helicity of the upconverted PL. They argue that it is the evidence for stimulated (nonradiative) exciton scattering from B- to A-states.

In terms of significance and potential impact, I find this work somewhere in the middle. The quality of the data is impressive, especially the narrow exciton linewidths. The upconversion effect is also interesting. However, this effect is probably of most interest to researchers in the specific field of TMD optics. I do not see it to be super- exciting for broader physics audience. As for the claim of stimulated scattering, it is plausible but lacks a solid proof. The authors did not demonstrate any control over this putative process.

I am undecided regarding whether this paper deserves publication in Nature Comm. In my opinion, it is a marginal case.

My final comment is about missing citations. In the 1st paragraph, the grouping [8-12] includes the follow-up work of Ref. 10 but not the original discovery [Butov et al, Nature 418, 751 (2002)]. In the second grouping [16-20] an important paper on exciton condensation [High et al, Nature 483, 584 (2012)] is overlooked. In the 2nd paragraph the grouping [42-52] includes Ref. 49 but not the earlier competing publication: Basov et al, Science 354, no. 6309 (2016), DOI: 10.1126/science.aag1992 .

Reply:

We thank the reviewer for this positive review and the precise analysis of our manuscript. "The quality of the data is impressive, especially the narrow exciton linewidths. The upconversion effect is also interesting."

The only critical remark concerns the impact of our work beyond the TMD material community. Here it is important to note that we think that bosonic physics of excitons previously observed in the systems with small binding energy is here transferred for the first time to a novel 2D system with large binding energy where, e.g., the Mott transition would require much higher densities. This leaves far greater room in experiments to explore exciton physics over a very broad range of densities. This is now suggested in the manuscript.

To further broaden the impact of our manuscript, we have also added the references suggested by the referee. The Nature papers by Butov *et al* [now Ref. 13] and High *et al* [now Ref. 22], are now added, the Science paper of Basov *et al* was already referenced in the initially submitted version [now Ref. 51].

Reviewer #2 (Remarks to the Author):

The authors report on low-temperature optical spectroscopy experiments on a monolayer of the TMDC WSe₂. In measurements where they illuminate the A exciton transition in their sample resonantly, they observe luminescence emission peaks at energies well above the laser energy, and identify these high-energy peaks as excited A exciton states and B exciton states. They demonstrate a strong sensitivity of this ‘upconverted’ luminescence on excitation laser energy, and a power dependence of the luminescence intensity which has about twice the slope of the 1s A exciton luminescence. The authors propose a mechanism of 2-photon absorption, with the 1s A exciton as a real intermediate state, leading to formation of B excitons from highly excited (continuum) states.

By using circularly polarized, resonant excitation of the 1s A exciton transition, they are able to generate significant circular polarization of the upconverted, excited A exciton state and B exciton emission. Remarkably, the circular polarization degree of these two luminescence peaks is opposite to each other, and the authors attribute the counter-polarized B exciton emission to B-exciton-A-exciton scattering, which is more probably between excitons in the same valley. This mechanism would lead to an imbalance of B exciton occupation in the different valleys, even though the B exciton formation is not valley-index-conserving. This mechanism is supported by density- dependent measurements clearly showing increasing counter-polarized luminescence due to increased exciton-exciton scattering.

The results presented in this manuscript are novel, and the claims made by the authors are well-supported by the experiments. They convincingly present an elegant way to establish a B exciton valley polarization in a TMDC, and their results are likely to invigorate further research of B exciton valley dynamics. More importantly, they provide clear evidence for valley-selectivity of bosonic scattering processes in a TMDC, which will have to be considered in future research aimed at studying exciton- exciton interaction and potential exciton condensation. Thus, the paper will most certainly be highly interesting to other groups working in this field.

Overall, the paper is very well-written, and the figures are clear and informative. While I would highly recommend publication of this manuscript in Nature Communications without major changes, I do have some questions which should be addressed by the authors, as well as some suggestions for minor revisions of the paper.

My questions are related to phenomena which are, so far, specific to the tungsten-based TMDCs: the conduction-band spin splitting in these materials makes the lowest-energy A exciton state a dark state, as demonstrated, among others, by some of the authors of the present work (NATURE COMMUNICATIONS | 6:10110). By contrast, the lowest-energy B exciton state (with the electron in the lower spin-split conduction band) is bright. Have the authors considered the potential impact of this splitting on the effects they observe? Might this actually limit the feasibility of the B exciton generation mechanism in the related TMDC MoSe₂?

The second question concerns the formation of biexcitons, which have been reported for WSe₂ under intense excitation. Do the authors have any indications of biexciton emission for their highly resonant A exciton excitation conditions?

Suggestions for minor changes:

- There is an additional feature above the 2s A exciton transition that is tentatively identified as the 3s A exciton – is the energy difference in agreement with the models for the excited states of TMDC monolayers?

- Related to the point above: a two-component fit is used to track the 2s A exciton and the higher-energy feature as a function of temperature, but this is not discussed in the caption of Fig.2d, and should be mentioned there. It also appears that there are significant changes of the linewidth, spectral weight and maybe also of the splitting between the two features. Given that these features have already been analyzed, it might be helpful to include the data from these fits in the supplement for interested readers.

Small comments:

- The authors mention the use of a pulsed laser in the methods. I assume that this was used to resonantly excite the B exciton transition for the PL spectra shown in Fig.1d? If that is the case, it should be mentioned in the caption of the Figure or in the text, or added within the figure itself. I would also suggest to indicate for which other measurements the pulsed excitation was employed,

- While the continuum states $|f\rangle$ are mentioned in reference to Fig. 2c both within the main text and the supplementary, the actual symbol $|f\rangle$ is not depicted in that figure.

Reply :

We thank the reviewer for this very positive review, the main points we tried to address are perfectly summarized in the review letter. We thank the reviewer to be strongly in favor of publication of this work. We address below point-by-point the comments and questions raised.

- *Molybdenum based TMDCs*

How the exciton scattering scenario changes in the related semiconductor ML MoSe₂ is an excellent question. We did not have the opportunity to do detailed experiments. We would need to verify if the upconversion mechanism is effective in ML MoSe₂. Here are the differences we would expect: As the ground state in MoSe₂ MLs is expected to be bright, it is not certain that a high exciton density can be reached, as excitons have a very short radiative lifetime. In ML WSe₂ the dark states below the bright states might actually be very useful as a long-lived reservoir for excitons, that helps to build up sufficient densities to observe scattering. Following the reviewer's questions, we now mention future experiments on ML MoSe₂ as a very interesting perspective of this line of research in the manuscript.

- *Possibility of biexciton formation*

Again this is a very good point, and with the well-separated lines we have in principle good visibility to see new peaks emerge. We have not found clear biexciton emission. First, we do not see any emerging emission at the energy reported for WSe₂ on SiO₂ by You *et al* in Nature Physics 11, 477 (2015). We did not observe any peak with a clear

superlinear or quadratic dependence of the emission intensity as a function of laser power. We are at the very beginning of understanding the influence of the dielectric environment on the Coulomb interaction, which will also impact biexciton binding energy and formation processes.

- *Excited A-exciton states labeled 2s and 3s*

We agree with the referee that this is an interesting point, but we are very careful to comment on the exact nature of these transitions, as more experiments are needed to really confirm their origin. In the work of Stier *et al*, Nano Letters 16, 7054 (2016) a strong reduction in exciton binding energy is suggested when encapsulating the samples. We would expect that also the excited exciton states are shifted, but precise predictions are difficult at the current level of understanding due to the strongly non-hydrogenic exciton series. As an alternative, what we label the 3s state could also be a phonon replica of the 2s. We use 2s and 3s here more as plausible labels, the exact nature is not critical for this paper more focused on scattering of B with A-excitons, but is of course an interesting subject for future studies. We have included the paper of Stier *et al* in the reference list [now Ref. 62] to make the reader aware of a possible impact of changing the dielectric environment on the exciton state energies.

As suggested by the reviewer, we have now plotted the temperature evolution of the peaks labeled 2s and 3s : the transition energy, linewidth and relative intensity ration. This extra figure is shown in the supplement (Fig. S4) and might help future studies on these excited states.

- *Use of pulsed lasers*

The reviewer is correct. The pulsed laser system is used for only one spectrum presented here in figure 1d: for resonant excitation of the B-exciton. This is now clarified in the text, as suggested by the reviewer. All the other experiments are performed with a low power, cw Ti-Sa laser.

- *Labeling the final state in Fig. 2c*

We have now labeled, as requested, the continuum, which is the final state, as $|f\rangle$ in figure panel 2c. Now the labels on the figure correspond to the discussion in the text.

Reviewer #3 (Remarks to the Author):

This paper delivers interesting and important result for the highly competitive field of TMDC materials, where the efforts of many leading groups in solid state physics are focused nowadays. The authors present very nontrivial experimental observation of the strong upconversion signal from B-exciton state, when the excitation is resonant for the A-exciton, which is lower in energy. Even more surprising is the fact that the emission of

B-exciton is circularly polarized when the circular polarized excitation is chosen, and the polarization degree is negative to the excitation one. By careful experimental checks the authors convincingly exclude few possible mechanisms and conclude that this polarization originates from the bosonic character of the exciton scattering. This is novel and principally important result for TMDC, other 2D materials, but also semiconductor microcavities.

Paper is very clearly written to be understandable for general reader. I recommend it for publication in Nature Communications.

I have only one question that author may wish to comment. Does encapsulation of WS₂ in boron nitride influence of the exciton parameters (e.g. binding energy), say via dielectric confinement?

Reply :

We thank the reviewer for taking the time to analyze our work in detail. We really appreciate that the reviewer considers our findings to be important beyond the TMDC community.

- *Influence of the dielectric environment*

The reviewer is raising a very interesting point. At first sight, it is surprising that the A-exciton transition in hBN encapsulated samples and samples simply exfoliated onto SiO₂ show almost the same emission energy. Here the community suggests that this is due to the almost exact compensation of two effects, namely:

- (i) the reduction of the exciton binding energy and
- (ii) the reduction of the quasi-particle bandgap.

as described for instance in Stier et al, Nano Letters 16, 7054 (2016). As a result, there is almost no overall shift of the emission when comparing encapsulated and non-encapsulated samples. Following the referees' question, this is now stated in the manuscript for the benefit of the reader and we quote the work of Stier et al as Ref. 62. We believe that tuning the exciton energies by adjusting the dielectric environment is an interesting idea for future experiments.

Reviewers' Comments:

Reviewer #2 (Remarks to the Author)

The authors have fully addressed all of the questions raised in my initial review and modified their manuscript accordingly. Thus, I highly recommend publication of the revised manuscript in its present form.

Reviewer #4 (Remarks to the Author)

I am fine with authors revisions and recommend publication in present form.

REVIEWERS' COMMENTS:

Reviewer #2 (Remarks to the Author):

The authors have fully addressed all of the questions raised in my initial review and modified their manuscript accordingly. Thus, I highly recommend publication of the revised manuscript in its present form.

Reviewer #3 (Remarks to the Author):

I am fine with authors revisions and recommend publication in present form.

Reply : We really appreciate that both reviewers accept this revised version in its present form. We have therefore not implemented any further changes to the manuscript and supplementary information.